# Quantification of speech and synchrony in the conversation of adults with autism spectrum disorder

**Keiko Ochi[1]\*, Nobutaka Ono[2], Keiho Owada[3], Masaki Kojima[3], Miho Kuroda[3], Shigeki Sagayama[4], Hidenori Yamasue[5]\***

**1** School of Media Science, Tokyo University of Technology, Hachioji, Japan, **2** Department of Computer Science, Graduate School of Systems Design, Tokyo Metropolitan University, Hino, Japan, **3** Department of Child Psychiatry, School of Medicine, The University of Tokyo, Tokyo, Japan, **4** University of Tokyo, Tokyo, Japan, **5** Department of Psychiatry, Hamamatsu University School of Medicine, Hamamatsu, Japan

\* ochikk@stf.teu.ac.jp (KO); yamasue@hama-med.ac.jp (HY)

**Data Availability Statement:** The data underlying the results presented in the study cannot be made publicly available due to ethical restrictions imposed by the the Ethical Committee of the University of Tokyo Hospital. The data are available

## Abstract

Autism spectrum disorder (ASD) is a highly prevalent neurodevelopmental disorder characterized by impairments in social reciprocity and communication together with restricted interest and stereotyped behaviors. The Autism Diagnostic Observation Schedule (ADOS) is considered a 'gold standard' instrument for diagnosis of ASD and mainly depends on subjective assessments made by trained clinicians. To develop a quantitative and objective surrogate marker for ASD symptoms, we investigated speech features including $F_0$, speech rate, speaking time, and turn-taking gaps, extracted from footage recorded during a semi-structured socially interactive situation from ADOS. We calculated not only the statistic values in a whole session of the ADOS activity but also conducted a block analysis, computing the statistical values of the prosodic features in each 8s sliding window. The block analysis identified whether participants changed volume or pitch according to the flow of the conversation. We also measured the synchrony between the participant and the ADOS administrator. Participants with high-functioning ASD showed significantly longer turn-taking gaps and a greater proportion of pause time, less variability and less synchronous changes in blockwise mean of intensity compared with those with typical development (TD) (p<0.05 corrected). In addition, the ASD group had significantly wider distribution than the TD group in the within-participant variability of blockwise mean of log $F_0$ (p<0.05 corrected). The clinical diagnosis could be discriminated using the speech features with 89% accuracy. The features of turn-taking and pausing were significantly correlated with deficits of ASD in reciprocity (p<0.05 corrected). Additionally, regression analysis provided 1.35 of mean absolute error in the prediction of deficits in reciprocity, to which the synchrony of intensity especially contributed. The findings suggest that considering variance of speech features, interaction and synchrony with conversation partner are critical to characterize atypical features in the conversation of people with ASD.

from the corresponding author (HY: yamasue@hama-med.ac.jp) or the Ethical Committee of the University of Tokyo Hospital (ethics@m.u-tokyo.ac.jp) on request from investigators providing a methodologically sound proposal and whose proposed use of the data has been approved by an independent review committee identified for this purpose. Maintenance of the identified data set will be ended 5 years following article publication, but the de-identified data will be maintained indefinitely.

**Funding:** Neither the funder nor sponsor, the Strategic Research Program for Brain Sciences from the Japan Agency for Medical Research and Development (JP18dm0107134 to HY), had any involvement in the data collection, analyses, writing, or interpretation of the study. This work was also partially supported by a JSPS KAKENHI Grant-in-Aid for Scientific Research (A) (Grant Number: 16H01735 to NO).

**Competing interests:** The authors have declared that no competing interests exist.

## Introduction

Autism spectrum disorder (ASD) affects approximately 1% of the general population [1], and currently there is no approved medication for the core symptoms. Individuals with ASD show deficits in social communication and interactions, including nonverbal communicative behaviors (e.g., eye contact, gestures, facial expressions, and speech prosody), as core symptoms [2]. Because diagnosis currently depends mainly on subjective assessment of these behaviors by trained clinicians, developing objective, quantitative, and reproducible assessments for social behavior is expected to improve the accuracy of diagnosis and to promote the further development of novel therapies by accurately detecting time-course changes in the severity of ASD core symptoms [3, 4].

The Autism Diagnostic Observation Schedule (ADOS) [5] is the gold-standard diagnostic tool for ASD. Using a semi-structured administration and scoring system, variability in assessment with ADOS is minimized across administrators and subjects. However, because ASD has been considered as static traits, available diagnostic tools, including ADOS, were not originally formulated to be repeatable and to detect changes in symptoms over time. We assume that the quantification of behavioral characteristics based on video footage recorded during easily repeatable activities in ADOS may provide such a valid and reliable quantitative measure for assessing longitudinal changes in core ASD symptoms.

A substantial body of literature describes the characteristics of speech prosody of people with ASD using both objective and subjective measures [6]. Although there are some discrepancies in previous findings [7–15], a meta-analysis revealed higher mean pitch and wider pitch variance in individuals with ASD [16]. These studies explored the significant differences in mean values between typically developing (TD) and ASD groups. However, Green and Tobin [17] showed that speakers with ASD can be classified into three groups with narrow, wide and typical pitch ranges, pointing out the possibility that comparison of mean value is not adequate. In addition, assessment criteria of ADOS include both exaggerated and monotonous prosody, or speech rate that is too fast or too slow, because these are considered characteristics of ASD.

Some studies have focused on conversation that is assumed to include the characteristics of interaction with other persons. For instance, in experiments measuring the timing of turn-taking, Heeman *et al.* reported that the turn-taking gap immediately after a question was significantly longer for children with ASD than for those with TD but not after an utterance [18].

In conversation, synchrony—a type of entrainment—is commonly observed in many languages [19]. It can be measured by tracking the similarity of the change trends of the two speakers [20]. Pérez and colleagues revealed that both synchrony and asynchrony play a role in entrainment [21]. In addition, although less reciprocal conversation is one of the core features of ASD in social communication [5], few studies have investigated engagement in conversation by individuals with ASD [22]. Regarding speech features, previous studies have attempted to use acoustic and prosodic features, including statistics related to fundamental frequency ($F_0$) [23], emotional expression [24], and turn-taking features [25, 26], although the conversations examined in these studies were confined to relatively simple interaction in children with ASD.

The aim of our study is to examine whether quantified speech features in recorded footage can be used to detect the behavioral characteristics of individuals with ASD compared with those with TD, and to develop an objective measure that facilitates the detection of changes over time via longitudinal assessments. We analyzed the speech of individuals with ASD and TD from two points of view: First, we hypothesized that the group with ASD would have a wider range of speech features than the TD group even if the distributions of the two groups

overlap. Second, we focused on social interaction where deficits in social communication are assumed to occur for people with ASD. Thus, we compared not only the mean values, but also the variance of the distribution of the speech features of the two groups. In addition, analysis of prosody was conducted with long sliding windows to measure the similarity of change trends with conversational partner, to describe the degree of synchrony of the two speakers.

Speech samples were collected from adults with ASD and TD during an activity from ADOS. We investigated speech features including pitch, intensity, speech rate, speaking time, and turn-taking gaps, where the mean and standard deviation (SD) were calculated as basic statistics for quantification. We also carried out a block analysis using an 8s long sliding window to quantify dynamic changes and synchrony with the conversational partner. *F*-tests were conducted prior to *t*-tests, comparing the ASD and TD groups to investigate the within-group variance of the two groups. After that, we performed discrimination analysis to classify the individuals into ASD or TD groups using the proposed speech features. The relationships between the speech features and ADOS scores were explored by correlation analysis. Regression analysis was employed to predict ADOS score from the speech features and determine the combination of features best-suited for the prediction.

## Methods

### Participants

Recruitment, clinical assessments, and data collection were conducted at the University of Tokyo Hospital. Sixty-five adult males with high-functioning ASD participated in the study. They had all been diagnosed with ASD, Asperger's syndrome, or an unspecified pervasive developmental disorder based on Diagnostic and Statistical Manual-Revision IV-Text Revision (DSM-IV-TR) [27]. A psychiatrist (HY) experienced in developmental disorders made a diagnosis of ASD based on the strict criteria of DSM-IV-TR. A certified psychologist (MK) confirmed the diagnosis using the Japanese version of the Autism Diagnostic Interview-Revised [28] and ADOS Module 4 [5]. The Wechsler Adult Intelligence Scale revised III (WAIS) [29] was used to confirm that the participants' full scale IQs were above 80.

Participants were confirmed not to fulfill the exclusion criteria, viz. 1) primary psychiatric diagnoses other than ASD; 2) instability in symptoms of comorbid mental disorders such as mood disorder or anxiety disorder; 3) history of changes in medication or medication dosage of psychotropics within one month of assessments; 4) currently under treatment with more than two different psychotropics; 5) currently under treatment for comorbid ADHD with atomoxetine or methylphenidate; 6) history of seizures or traumatic brain injury with loss of consciousness for longer than 5 minutes; and, 7) history of alcohol-related disorder, substance abuse or addiction.

In addition, 17 adult Japanese males with a history of typical development (TD) were recruited and matched for age, parental socioeconomic background [30], handedness, and intellectual level. The verbal IQs of these participants were estimated using the Japanese version [31] of the National Adult Reading Test (NART) [32]. While the NART can estimate the verbal IQs of TD individuals, this can be problematic for those with ASD because of imbalanced intellectual abilities often associated with ASD, and well-known discrepancies between subscales of the WAIS. Therefore, the IQs of the participants with ASD were assessed using the WAIS-III. TD participants were screened by trained psychiatrists (HY and KO) for the following exclusion criteria: presence and/or past history of neuropsychiatric disorders using the Structured Clinical Interview [33] and family history of neuropsychiatric disorders in their first-degree relatives.

The Ethical Committee of the University of Tokyo Hospital approved this study (10245). After a complete explanation of the study, participants' mental capacity to consent was confirmed by a psychiatrist (H.Y. or K.O.), and written informed consent was obtained from all participants.

## ADOS administration

ADOS module 4 was administered to all participants by a single administrator (HY), who had completed a training course on the use of ADOS for research. Administration was validated by a certified administrator (MK). All ADOS administrations were recorded on video, with scoring the participants with ASD verified by a single certified administrator (MK), thus minimizing inter-administrator and -rater variability. We used the recorded videos for activity 7, 'Emotions' in ADOS module 4, for analysis, because participants were required to participate in a conversational interview with the administrator's fixed questions about various types of emotional experiences, likely to reveal a variety of affective prosody.

From the 65 participants with ASD, two speech samples were excluded because the audio recordings were incomplete. In addition, one participant with ASD withdrew consent and his data were then excluded. In total we analyzed 79 recordings from 62 participants with ASD and 17 individuals with TD.

## Recording paradigm

A pair of wireless lavalier microphones (ECM-AW4, Sony Corporation, Tokyo, Japan) was attached to the collars of the participant and the ADOS administrator, to record their conversation with a favorable signal-to-noise ratio (SNR), without disrupting their natural speech like headset microphones sometimes do. The sampling frequency was 48 or 44.1 kHz, downsampled to 16 kHz after recording and with quantization with 16-bit precision.

## Speech feature processing

We extracted two types of speech features related to prosody and voice activity. Preprocessing for both features involved manual extraction of the inter pausal units (IPUs) of each participant and administrator. IPUs were defined as speech intervals divided by silence longer than 200 ms and not included within a word.

**Prosodic features.** Prosodic features analyzed in this study included the log $F_0$, intensity, and speech rate. Both the log $F_0$ and intensity were computed using Praat [34], with a frame shift of 10 ms. This was only done for spoken parts of the detected IPUs of each participant or administrator. Overlaps in participant and administrator speech were excluded from the analysis. For robust $F_0$ extraction, we followed the method described in [35]. We first applied Praat pitch detection with a fixed pitch floor and ceiling for all participants. The pitch ceiling and floor were set at 32 and 200 Hz, respectively, based on observation of the speech data. For the distribution of the extracted pitch for each participant, we applied Praat pitch detection again with the pitch floor set to 0.75 times the lower quantile and the pitch ceiling set to double the upper quantile. The pitch floor and ceiling were manually tuned for only two participants. In the intensity analysis, the frame length was 32 ms.

Finally, the IPUs were transcribed as phoneme sequences using their temporal boundary information by the speech recognition engine Julius. A Japanese monophone Hidden Markov Model (HMM) distributed in the Julius dictation kit (version 4.0) [36] was used as an acoustic model. The number of morae in each IPU was obtained from the transcription to calculate the speech rate. In this study, speech rate was calculated by dividing the total number of morae by total duration of speech excluding silence, so that the pause and 'rate of articulation' were

analyzed separately. Note that the mora was used as the unit of calculation for speech rate instead of the syllable, because morae are the rhythmic units in Japanese.

Let $f_0[i]$ and $J[i]$ be the value of $F_0$ (in Hz) and intensity (in decibels) of a participant at the $i$ th frame. We calculated the $F_0$ and intensity only within IPU, and $F_0$ was extracted only at the voiced frame. We then calculated the overall and blockwise statistics.

**1) Overall session features.** For the overall session statistics, we calculated the mean and SD for $F_0$ and intensity over the whole session. The mean of the intensity was excluded since it could depend on the unknown distance between the microphone and the speaker. The mean of log $F_0$, mean of intensity, SD of log $F_0$, and SD of intensity were extracted as follows:

$$m_{f_0} = \frac{1}{\#\Gamma_V} \sum_{i \in \Gamma_V} \log f_0[i], \tag{1}$$

$$s_{f_0} = \sqrt{\frac{1}{\#\Gamma_V} \sum_{i \in \Gamma_V} (\log f_0[i] - m_{f_0})^2}, \tag{2}$$

$$m_J = \frac{1}{\#\Gamma} \sum_{i \in \Gamma} J[i], \tag{3}$$

$$s_J = \sqrt{\frac{1}{\#\Gamma} \sum_{i \in \Gamma} (J[i] - m_I)^2}, \tag{4}$$

Where $\Gamma_V$ and $\Gamma$ are the set of frame indices when $F_0$ and intensity were calculated, respectively.

**2) Blockwise Features.** Fig 1 shows the flow for calculation of blockwise statistics from frame features. To analyze the long duration changes and synchrony of the pitch and intensity, we performed a block analysis using the time-aligned moving average (TAMA) method [37]. We defined $x[i]$ as a value of the log $F_0$ or intensity at the $i$th frame. The blockwise mean value of $k$th block is defined by:

$$x_B[k] = \frac{1}{\#\Gamma_B[k]} \sum_{i \in \Gamma_B[k]} x[i]. \tag{5}$$

where $\Gamma_B[k]$ is a set of frame numbers included in the $k$th block. It is defined by:

$$\Gamma_B[\text{k}] = i \mid \frac{n}{2} k \leq i \leq \frac{n}{2}(2k+1) \wedge x[i] \text{ is calculated at the } i\text{th frame} \}, \tag{6}$$

where $n$ is the number of frames included in a block. Note that $\Gamma_B[k]$ equals the numbers of the frames included in IPUs in the case of intensity, whereas $\#\Gamma_B[k]$ is the number of voiced frames in that block in the case of $F_0$.

The SD of the blockwise mean of the log $F_0$ or intensity is similarly obtained by:

$$s_B = \sqrt{\frac{1}{N} \sum_{k=1}^{N} (x_B[k] - m_{x_B})^2} \tag{7}$$

where $N$ and $m_{x_B}$ are the number of blocks that comprise the whole session and the mean

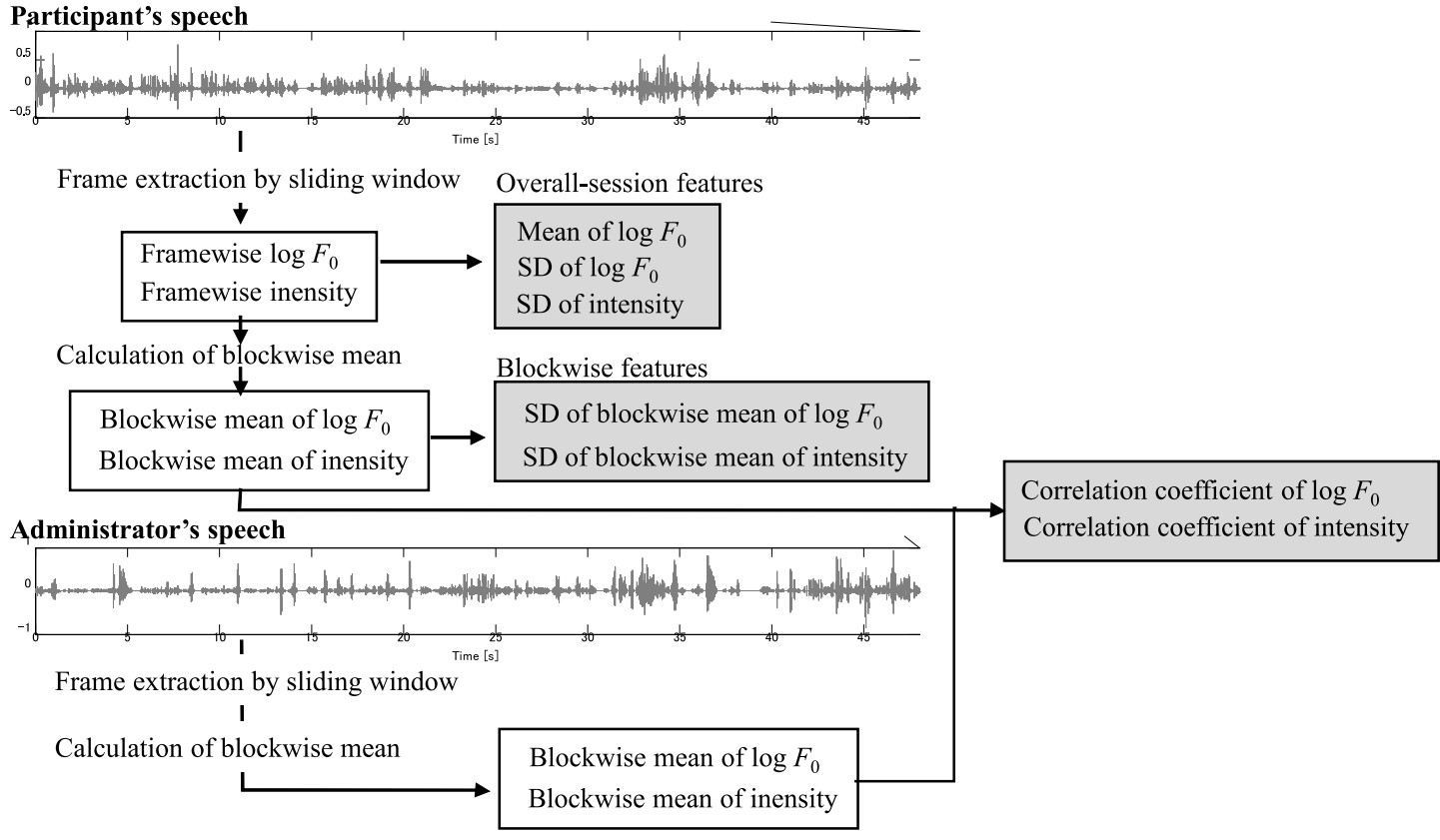

**Fig 1. Calculation of frame and block statistics of prosodic features.**

value of $x_B[k]$ over the whole session, respectively.

$$m_{x_B} = \frac{1}{N}\sum_{k=1}^{N} x_B[k] \tag{8}$$

The normalized correlation coefficient was calculated using Eq 9. This value represents the similarity of the prosodic features of a participant and administrator in the activity

$$r_{xy} = \frac{\sum_{k=1}^{N}(x_B[k] - m_{x_B})(y_B[k] - m_{y_B})}{\sqrt{\sum_{k=1}^{N}(x_B[k] - m_{x_B})^2 \sum_{k=1}^{N}(y_B[k] - m_{y_B})^2}} \tag{9}$$

where $x_B[k]$ and $y_B[k]$ are the blockwise mean of a prosodic feature of a participant and an administrator at $k$th block.

The block length and step were set to 16s and 8s, respectively, in this study. Overlapping speech of an administrator and participant was excluded from the analysis. We then obtained the SD of the average intensity and pitch in every block (blockwise mean) of a participant, and the correlation coefficient for an administrator and participant.

**Voice activity features.** Voice activity features were obtained from the detected IPUs. Speaking time was defined as the duration of the IPU in seconds. Fig 2 shows the definition of a turn-taking gap, viz. the time from the end of the administrator's utterance to the beginning of the participant's next utterance. When the participant started to speak before the administrator had finished speaking, the turn-taking time has a negative value. We did not include the

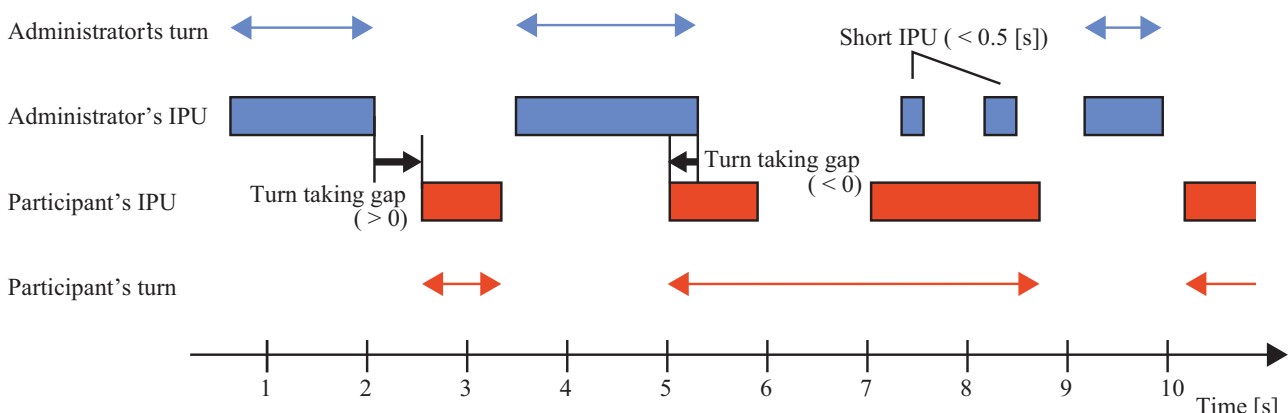

**Fig 2. IPUs and turn-taking gaps.**

backchannels, which are defined as the utterances inserted into the interlocutor's utterances without interrupting in many literatures [38], into a turn based on the study by Sato *et al.* [39]. For the calculation of turn-taking gaps, we omitted utterances that overlapped entirely with the administrator's utterance, regarding them as backchannels or failures of turn-taking in which the participant continued to speak after the administrator tried to take a turn.

We calculated the pause-to-turn ratio as the ratio of the total pause duration to the total speaking-turn duration in a session. For automatic processing, we simply defined the speaking turn as the time segment from the beginning of an utterance longer than the threshold to the end of the utterance before the start of an utterance from the other speaker that exceeded the threshold. We heuristically set 0.5s as the threshold, which means that utterances shorter than 0.5 seconds, such as backchannels, did not count as turns. The pause-to-turn ratio was large when a participant maintained a long silence after obtaining a turn.

**Statistical analysis.** To assess the normality of the distribution of the speech features of the ASD and TD group, we conducted Kolmogorov-Smirnov tests on each feature of each group. Based on the results which showed the non-normality in the distribution of the mean and SD of turn-taking gap and the turn-to-pause ratio ($p < 0.05$), we took the logarithm transform on the three features, referring to the findings that the logarithm of the duration of the gaps and overlaps of turn takings have a normal distribution [40] [41]. For only the mean of turn-taking gap, we added a constant to all samples such that the minimum value of the feature was equal to 1 prior to the logarithm transform for non-negativity. We confirmed that Kolmogorov-Smirnov test showed no significance on these logarithmically transformed features.

Difference in the mean value of each feature between the ASD and TD groups was compared by *t*-test. Prior to the *t*-tests, we conducted *F*-tests to compare the variance because the features vary among individuals with ASD. Statistical significance for all analyses was set at $p < 0.05$ after adjustment with the Benjamini-Hochberg (B-H) multiple comparisons procedure [42].

As an exploratory analysis, we applied a support vector machine (SVM) to discriminate the two groups of speakers using the e1071 [43] package in the statistical environment R. We used linear kernel because it provided the best performance in our experiment. We evaluated the classifier by one-leave-out cross-validation. Because the sample sizes of the two groups were imbalanced, we weighted the objective function used in the training of the SVM according to the inverse number of the sample size of each class to avoid the undervaluation of the false negative. We tested three feature sets which consisted of all 13 speech features (Setting D1), only

features where results of $F$- or $t$-tests showed a significant diagnostic difference (Setting D2), and the best combination of features selected based on accuracy from all possible 8191 combinations (Setting D3).

The number of the possible combination (8191) represents the number of all cases where each of 13 speech features were used or not ($2^{13} - 1$). In the evaluation of the D3 setting, we used leave-one-out cross-validation to avoid overfitting. In details, the accuracy of each combinations of speech features (Combination $i$) was evaluated by the following equation:

$$a_i = \frac{\sum_{k=1}^{N_{\text{ASD+TD}}} b_k}{N_{\text{ASD+TD}}} \tag{10}$$

where $b_k$ takes 1 or 0 when the classification is accurate or not respectively, in the test for the $k$th participant's data by the classifier trained using all the other participants' data (i.e., all 78 participants except for the $k$th one). $N_{\text{ASD+TD}}$ represents the total number of the participants of the ASD and TD groups (79 in this study).

The correlational analyses of the extracted speech features and ADOS scores were conducted after omitting the features which had no significant differences in the $t$- and $F$-tests comparing the two groups. We focused on the score of the reciprocity in the ADOS score because it was assumed to reflect the attributes of the interaction between the participants and administrator. We also analyzed the correlation between the speech features and the scores of communication and repetitive domains of ADOS as references.

In addition to the above calculations, we considered the possibility that the speech features of the participants with ASD may have a wider distribution than those with TD. High ADOS scores arise when individuals' speech features are very great or small. Thus, in this study, we applied a transformation to speech features that showed significance in the $F$-test and no significance in the $t$-test, using the following equation to compute the distances from the mean of the speech features in the TD group:

$$\hat{x}_i = |x_i - m_{TD}| \tag{11}$$

where $x_i$ and $m_{TD}$ are the value of a speech feature of $i$th participant and the mean value of the speech feature in the TD group, respectively. These transformed speech features were used in the correlation analysis instead of the raw values. We used a permutation test for the SD of blockwise mean of log F0, which did not have a normal distribution (Kolmogorov-Smirnov test, $p < 0.05$).

In addition to the above analyses and experiment, we conducted a prediction experiment using the SVR included in the e1071 package in R with linear kernel. The following three feature sets were tested:

1. All 13 features (Setting R1)

2. The features included in Setting D2 that had a significant correlation with each ADOS score in the correlation analysis (Setting R2)

3. The features included in Setting D3 in the discrimination analysis (Setting R3)

4. The combination of speech features when root mean square error (RMSE) was the smallest (Setting R4).

The regressors in these four settings were evaluated based on the correlation coefficient of the predicted and rated ADOS score, RMSE, and mean absolute error (MAE), using a leave-one-out correlation coefficient. Similar to the Setting D3 in the discrimination analysis, the possible 8191 combination was evaluated in Setting R4. Carrying out one-leave-out cross-

validation in the same way, the correlation coefficients, MAE, and RMSE were obtained by averaging the values when the data a single participant was tested from the regressor trained by the data of the other participants, respectively.

## Results

### Background information

There was no significant difference in age, verbal IQ, or parental socio-economic status (SES) between the participants with ASD and those with TD ($p > 0.05$) (Table 1). The individuals with ASD had significantly lower self SES compared with the participants with TD ($p = 4.67 \times 10^{-14}$).

### Comparisons of groups with and without ASD (Table 2)

Fig 3 shows two examples of the time series of the mean intensity of the two speakers. In the upper case, the intensity values sometimes move in opposite directions and at other times together. In the middle case, the intensity of the participant moves inversely to that of the administrator. The lower case, in contrast, shows the intensity of the participant increases and decreases together with the administrator's value.

Table 2 shows the means and SDs of the two groups and the $p$-values for the $t$- and $F$-tests. Fig 4 shows the histograms of the log mean of turn-taking gap, the SD of blockwise mean of intensity, and the SD of blockwise mean of log $F_0$. Both $t$- and $F$-tests showed significant differences between the ASD and TD group in the log mean of turn-taking gap, and the log SD of turn-taking gap (adjusted $p<0.05$) (Table 2, Fig 4). For each of these three features, the means and SDs were significantly greater in participants with ASD than those with TD. Fig 4 shows the histograms of speech features that showed significance in $t$-and/or $F$- tests.

**Table 1. Demographic characteristics of participants with autism spectrum disorder (ASD) and those with typical development (TD).**

| Variable | ASD (N = 62) | | TD (N = 17) | | P-value |
|---|---|---|---|---|---|
| | Mean | SD | Mean | SD | |
| Age (range) | 26.9 | 7.0 | 29.6 | 7.0 | 0.05 |
| Socioeconomic status [1] | 2.8 | 1.2 | 1.2 | 1.2 | $4.67\times10^{-14}$ |
| Parental socioeconomic status [1] | 2.1 | 0.5 | 1.9 | 0.5 | 0.12 |
| Full-scale IQ [2] | 106.4 | 14.3 | | | |
| Verbal IQ [2] | 113.9 | 14.8 | 117.7 | 14.8 | 0.12 |
| Performance IQ [2] | 94.9 | 15.7 | | | |
| ADI-R Reciprocal social interaction | 21.9 | 16.3 | | | |
| Communication | 5.4 | 5.0 | | | |
| Restricted, repetitive, and stereotyped patterns of behavior | 3.7 | 2.3 | | | |
| ADOS Reciprocal social interaction | 8.9 | 2.1 | | | |
| Communication | 4.2 | 1.4 | | | |
| Stereotyped behaviors and restricted interests | 2.0 | 1.3 | | | |

[1]Assessed using the Hollingshead two-Factor Index of Social Position [30], in which a higher score indicates a lower status.

[2] The IQs of participants with ASD were measured using the Wechsler Adult Intelligence Scale. The verbal IQ of those with TD was estimated using the Japanese version of the National Adult Reading Test.

Abbreviations: IQ, intelligence quotient; ADI-R, Autism Diagnostic Interview-Revised; ADOS, Autism Diagnostic Observation Schedule; SD, standard deviation.

**Table 2. Means and SDs of speech features of each group and the adjusted p values.**

| Feature | Mean (SD) | | t-test | F-test |
|---|---|---|---|---|
| | ASD | TD | p | p |
| Mean of log $F_0$ | 4.8 (0.16) | 4.7 (0.15) | 0.26 | 0.78 |
| SD of log $F_0$ | 0.2 (0.075) | 0.19 (0.046) | 0.63 | 0.12 |
| SD of blockwise mean of log $F_0$ | 0.088 (0.042) | 0.089 (0.023) | 0.96 | 0.047* |
| Corr. of blockwise mean of log $F_0$ | 0.086 (0.25) | 0.23 (0.25) | 0.084 | 0.86 |
| SD of intensity [dB] | 5.9 (0.82) | 5.8 (0.62) | 0.71 | 0.38 |
| SD of blockwise mean of intensity [dB] | 2.6 (1.1) | 3.5 (0.77) | 0.0020* | 0.22 |
| Corr. of blockwise mean of intensity | 0.18 (0.24) | 0.34 (0.20) | 0.035* | 0.68 |
| Speech rate [mora/s] | 8.8 (0.55) | 8.9 (0.66) | 0.71 | 0.47 |
| Mean of speaking time [s] | 1.5 (0.49) | 1.6 (0.39) | 0.43 | 0.47 |
| SD of speaking time [s] | 1.2 (0.46) | 1.5 (0.47) | 0.11 | 0.86 |
| Log mean of turn-taking gap [s] | 0.52 (0.37) | 0.18 (0.15) | $3.8 \times 10^{-6}$* | 0.0023* |
| Log SD of turn-taking gap [s] | 0.0091 (0.76) | -0.35 (0.35) | 0.025* | 0.0099* |
| Log pause-to-turn ratio | -2.9 (1.1) | -3.5 (0.77) | 0.025* | 0.34 |

*adjusted $p < 0.05$

Regarding the SD and correlation coefficient of blockwise mean of intensity, the pause-to-turn ratio, the t-test found significant differences between ASD and TD groups, although the F-test found no significant difference. The SD of blockwise mean of intensity was significantly smaller in the individuals with ASD ($M = 2.6$, $SD = 1.1$) compared with those with TD ($M = 3.5$, $SD = 0.77$) (Fig 4b), although the SDs of intensity within the activity were not significantly different. A small proportion of the ASD group (6.5%) with high SD of blockwise mean of intensity ($> 0.2$), murmured or whispered the question asked by the administrator to themselves while they were thinking, e.g. Administrator: "What do you think is scary?" Participant: "Scary. . ."). These quiet repetitions expanded the range of intensity.

The correlation coefficient of the intensity of the individuals with ASD ($M = 0.18$, $SD = 0.24$) was also significantly smaller than those with TD ($M = 0.34$, $SD = 0.20$) (Fig 4c). The pause-to-turn ratio of those of ASD was significantly larger than that of ASD. The F-test for the blockwise mean of $F_0$ showed a significant difference, even though no significant difference was observed in the t-test.

All other speech features showed no significant differences in either t- or F-tests. Although there was no significant difference in the features other than the SD of blockwise mean of log $F_0$, the log mean and SD of turn-taking gap, and the pause-to-turn ratio in F-tests, the variance of all features except the speech rate and correlation coefficient of log $F_0$ were greater in the individuals with ASD than those with TD.

## Discrimination of ASD and TD groups (Table 3)

Table 3 shows the confusion matrix of the discrimination analyses in Settings D1, D2 and D3. Table 4 gives the accuracy and F-measure in each setting. Setting D3, with the best set of features, achieved 88.6% accuracy and 92.3% F-measure. The accuracy in Setting D2, which used the speech features selected based on t- and F-tests, did not exceed that of Setting D1. The accuracy of Setting D2 was 11.4% points below that of Setting D3.

Table 5 lists the selected features in Setting D3, and the accuracy and F-measure when each feature was used alone for the classification using SVM. The log mean turn-taking gap provided the highest accuracy when it was used alone for the classification. The log mean of turn-

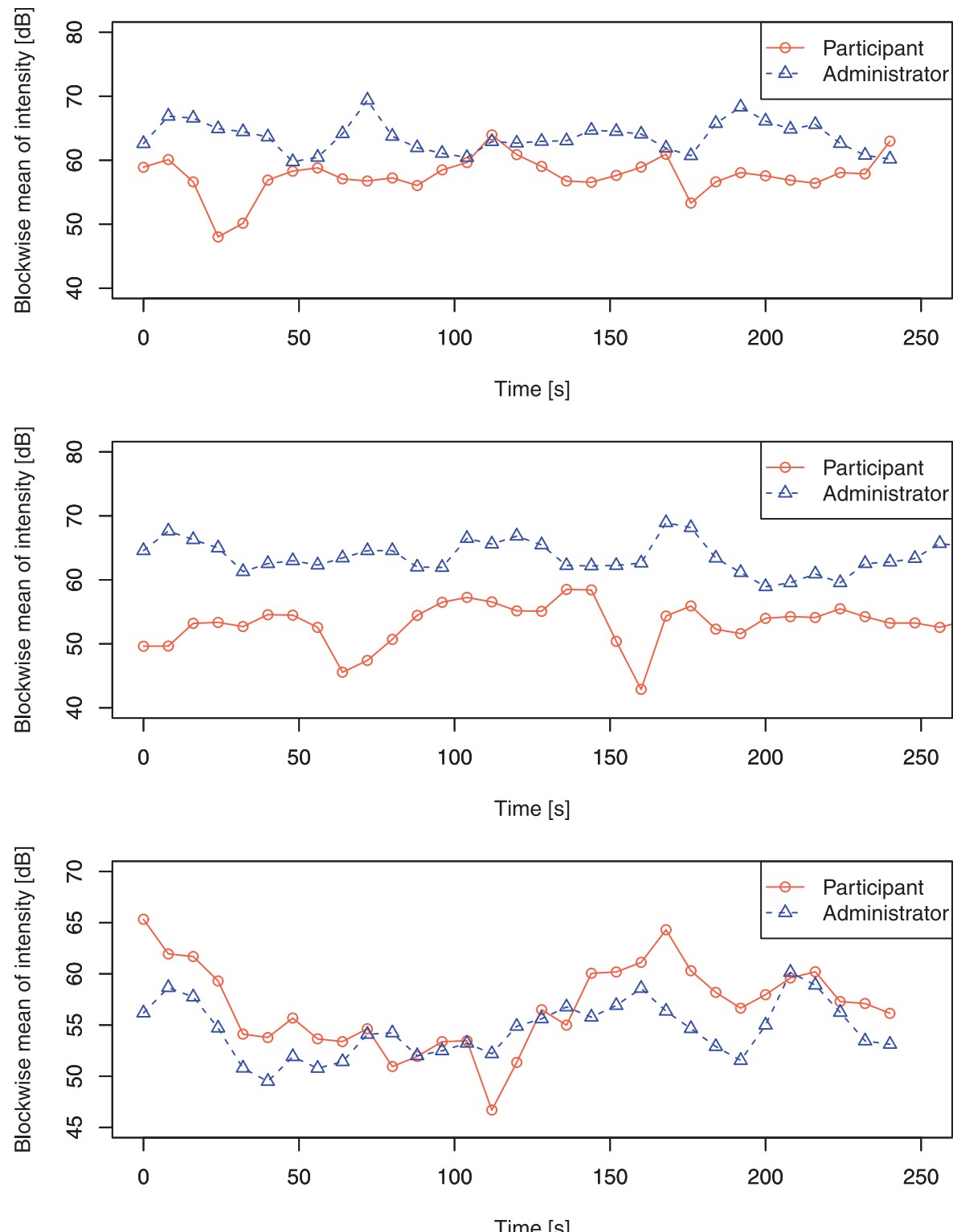

**Fig 3. Examples of time series of the blockwise mean of the intensity of the utterances of a participant with ASD and administrator (upper and middle panel) and the utterances of a participant with TD and administrator (lower panel).** The correlation coefficient of the features between the two speakers had a close-to-zero value (0.018) in the upper case, a negative value (-0.26) in the middle case, and a positive value (0.65) in the lower case.

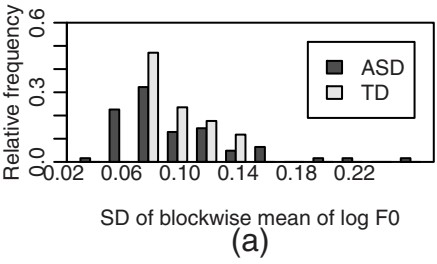
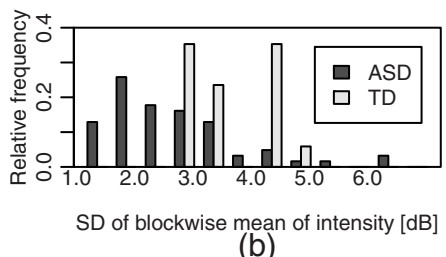

(a)
(b)

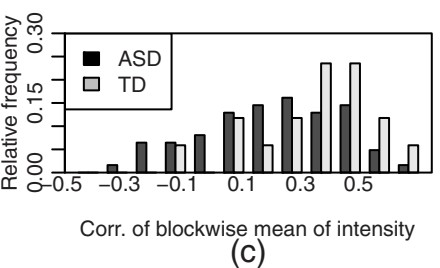
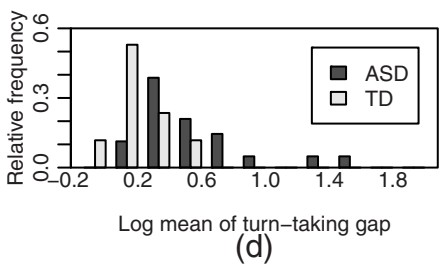

(c)
(d)

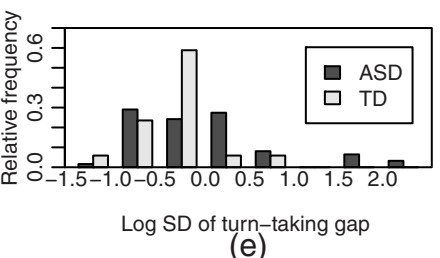
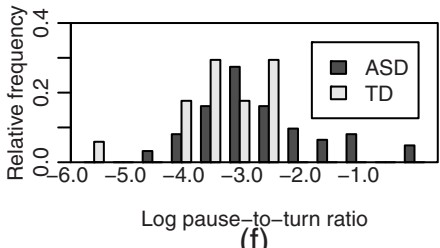

(e)
(f)

**Fig 4. Histograms of (a) SD of blockwise mean of log $F_0$, (b) SD of blockwise mean of intensity, (c) correlation coefficient of blockwise mean of intensity, (d) log mean of turn-taking gap, (e) log SD of turn-taking gap, and (f) log pause-to-turn ratio.** Nineteen out of 62 individuals with ASD (30.1%) had a mean turn-taking gap greater than 0.8s, whereas none of the 17 participants with TD had such a long mean gap. The pause-to-turn ratio of 17 individuals with ASD (27.4%) was greater than 0.1 whereas that of the individuals with TD was distributed in the range 0–0.1. Fourteen participants (22.6%) with ASD had negative correlation coefficients for intensity, whereas only one (5.9%) participant with TD had that. The absolute value of the correlation coefficient was less than 0.1 in 13 individuals with ASD (20.1%) and two with TD (11.8%).

taking gap and the SD of blockwise mean of intensity were included in at least the top 20 sets with accuracy, from all 8191 combinations.

## Correlations between speech features and ADOS score

Table 6 shows Pearson's correlation coefficients between ADOS scores and the speech features that showed significant differences in the $F$- or $t$-tests for individuals with ASD and TD

**Table 3. Confusion matrix of ASD and TD discrimination.**

| | Setting D1 | | Setting D2 | | Setting D3 | |
|---|---|---|---|---|---|---|
| Correct | Predicted | | Predicted | | Predicted | |
| | ASD | TD | ASD | TD | ASD | TD |
| ASD | 53 | 9 | 47 | 15 | 57 | 5 |
| TD | 11 | 6 | 9 | 8 | 3 | 14 |

**Table 4. Percentage of accuracy and F-measure in the discrimination analysis.**

|  | Setting D1 | Setting D2 | Setting D3 |
|---|---|---|---|
| Accuracy | 74.6 | 69.6 | 89.9 |
| F-measure | 84.1 | 79.7 | 93.4 |

**Table 5. Selected speech features in Setting D3 and the percentage of accuracy and F-measure when each feature was used alone for classification.**

| Speech feature | Accuracy |
|---|---|
| Mean of turn-taking gap [s] | 79.7 |
| Log SD of turn-taking gap | 67.0 |
| SD of blockwise mean of intensity [dB] | 64.6 |
| Corr. of blockwise mean of log F0 | 59.5 |
| Mean of turn-taking gap [s] | 79.7 |

(S1 Fig). The reciprocity scores showed significant correlations with the log mean of turn-taking gap ($r = 0.44$), the log of SD of turn-taking gap ($r = 0.35$) and the log of pause-to-turn ratio ($r = 0.41$) (adjusted $p<0.01$). In contrast, the communication and repetitive scores had no significant correlation with any speech features. Fig 5 shows scatter plots representing relationships of the log mean of turn-taking gap (left), log of SD of turn-taking gap (middle), and log of pause-to-turn ratio (right) to reciprocity score in the participants with ASD.

Although the SD of blockwise mean of intensity had no significant correlation with reciprocity, the scatter plot had a U- or V-shaped distribution. The reciprocity score has a low value when the feature is around the mean value of the TD. The reciprocity score increases as the features depart from the mean value of TD. The features of the participants with ASD were intensively distributed within the lower values: the features of 53 participants with ASD (85.5%) were lower than the mean for the TD group. It is also notable that the individuals whose value was distant from the average of TD received a relatively low score (e.g. 6). In the following regression analysis, the SD of blockwise mean intensity was used without converting because it did not improve the result.

## Regression analysis between speech features and ADOS score

Table 7 shows the correlation coefficients for the ADOS scores rated clinically and those predicted by the regressor, the MAE and RMSE of the predicted scores. The correlation of

**Table 6. Correlation coefficients (r) between the speech features and ADOS scores and adjusted p-values of correlation test.**

|  | Reciprocity | | Communication | | Repetitive | |
|---|---|---|---|---|---|---|
|  | r | adjusted p | r | adjusted p | r | adjusted p |
| SD of blockwise mean of log $F_0$ | n.s. | 0.84 | n.s. | 0.85 | n.s. | 0.26 |
| SD of blockwise mean of intensity [dB] | n.s. | 0.20 | n.s. | 0.33 | n.s. | 0.58 |
| Corr. of blockwise mean of intensity | n.s. | 0.27 | n.s. | 0.86 | n.s. | 0.43 |
| Log mean of turn-taking gap [s] | 0.41 | 0.0032* | n.s. | 0.33 | n.s. | 0.43 |
| Log of SD of turn-taking gap | 0.35 | 0.0096* | n.s. | 0.53 | n.s. | 0.43 |
| Log of Pause-to-turn ratio | 0.41 | 0.0032* | n.s. | 0.46 | n.s. | 0.43 |

*adjusted $p<0.05$

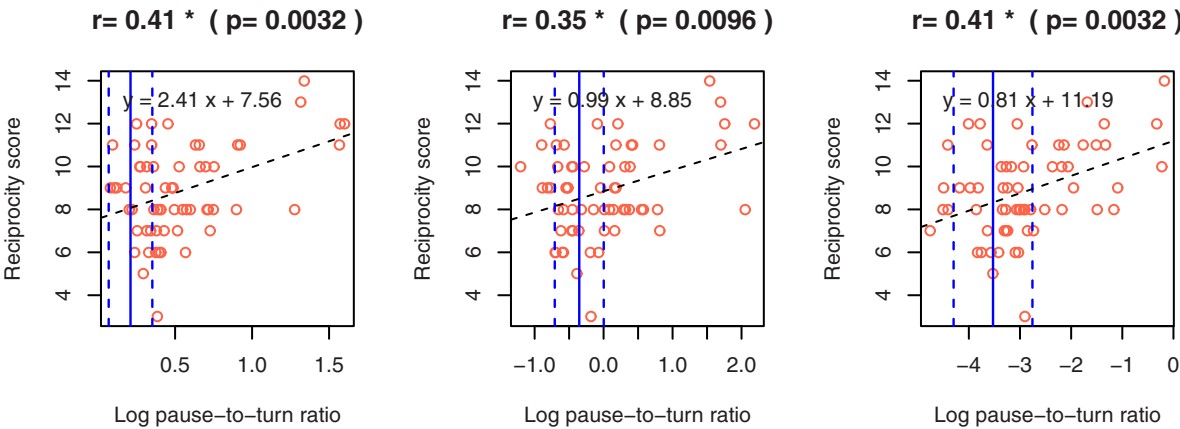

**Fig 5. Scatter plots between the features that have significant correlations with the reciprocity score.** The black dotted line represents a regression line. The blue solid and dashed lines represent the mean and mean ± SD of the feature among the participants with TD, respectively. A positive correlation was observed between the log of pause-to-turn ratio and reciprocity score of the individuals with ASD. At the same time, the distributions of the ASD and TD groups overlapped in the area between −5 and −2. Similar overlaps were observed in mean and log of SD of turn-taking gap.

reciprocity scores was improved in Setting R2 and R3 compared with Setting R1. In Settings R1, R3 and R4, the correlation between the predicted and rated reciprocity score was highest among three score domains.

Table 8 shows the selected speech features in Setting R4. The speech features selected for the prediction of communication and repetitive score was respectively involved in the selected features for the prediction of reciprocity. Log of mean of speaking time gave the highest correlation between rated and predicted reciprocity score when it was used alone. Using only the correlation coefficient of blockwise mean of intensity alone, the predicted reciprocity score did not yield a correlation with the rated score. However, the correlation between rated and predicted score was the third most degraded (0.47 from 0.58) when the feature was excluded from the best feature set.

Fig 6 shows the scatter plot of each category of ADOS scores predicted by SVR in Setting R4 and rated by the administrator. For the repetitive score, all of the predicted scores were in the narrow range between 0.95 and 3.5 although the rated scores ranged from 0 to 6.

**Table 7. Correlation coefficients (r) between rated and predicted ADOS scores, MAE and RMSE of predicted ADOS scores.**

| Domain | | Setting R1 | Setting R2 | Setting R3 | Setting R4 |
|---|---|---|---|---|---|
| Reciprocity | *r* | 0.42 | 0.32 | 0.26 | 0.58* |
| | MAE | 1.53 | 1.65 | 1.67 | 1.23 |
| | RMSE | 2.02 | 2.02 | 2.10 | 1.73 |
| Communication | *r* | 0.14 | - | 0.27 | 0.49* |
| | MAE | 1.34 | - | 4.62 | 0.96 |
| | RMSE | 1.60 | - | 4.85 | 1.21 |
| Repetitive | *r* | -0.34 | - | -0.14 | 0.18* |
| | MAE | 1.46 | - | 1.10 | 0.94 |
| | RMSE | 1.79 | - | 1.38 | 1.23 |

*Statistically significant.

**Table 8. Selected features in Setting R4 and the correlation coefficients between the rated and predicted ADOS scores when each feature was used alone (Corr. Single) and omitted from the best set (Corr. Omitted).**

| Domain | Features | Corr. Single | Corr. Omitted |
|---|---|---|---|
| Reciprocity | Mean of log F0 | -0.11 | 0.56 |
| | SD of blockwise mean of log F0 | -0.28 | 0.52 |
| | SD of intensity | 0.22 | 0.56 |
| | SD of blockwise mean of intensity | 0.042 | 0.49 |
| | Corr. of blockwise mean of intensity | 0.0020 | 0.48 |
| | Speech rate [mora/s] | -0.051 | 0.58 |
| | Log mean of speaking time | 0.46 | 0.11 |
| | Log mean of turn-taking gap | 0.37 | 0.43 |
| | Log pause-to-turn ratio | 0.17 | 0.52 |
| Communication | SD of blockwise mean of intensity | 0.077 | 0.45 |
| | Speech rate [mora/s] | -0.076 | 0.38 |
| | Log mean of speaking time | 0.40 | 0.24 |
| | Log SD of speaking time | 0.37 | 0.47 |
| | Log Mean of turn-taking gap | 0.16 | 0.46 |
| | Log pause-to-turn ratio | 0.044 | 0.31 |
| Repetitive | Mean of log F0 | 0.016 | 0.14 |
| | SD of blockwise mean of log F0 | 0.16 | 0.0031 |
| | Log mean of speaking Time | 0.51 | 0.18 |

## Discussion

The current study showed that individuals with ASD: (1) spoke with a uniform volume regardless of topics; (2) used wide variations in pitch depending on circumstances; (3) were less synchronous with interlocutors in volume of voice; and (4) demonstrated longer pauses between and within turns (Table 2, Fig 4). It was further noted that: (5) the length of pauses correlated with the reciprocity score in ADOS (Fig 5); and (6) the discrimination and regression analysis could successfully detect ASD individuals and predict the reciprocity score (Tables 3–7, Fig 6).

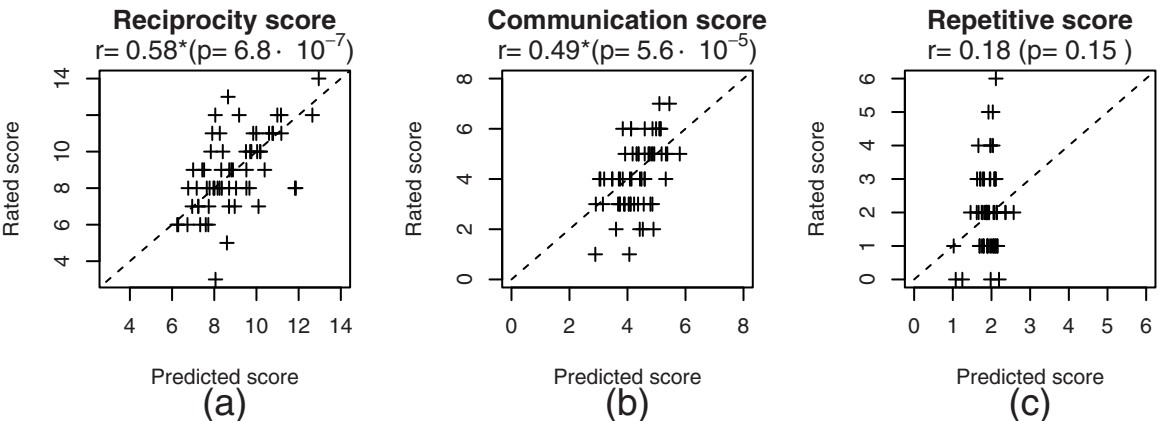

**Fig 6. Scatterplots of the (a) reciprocity, (b) communication and (c) repetitive scores that were predicted in Setting R4 and rated by the administrator.**

## Comparison of groups with and without ASD

**Variance of within-participant blockwise statistics.** The SD of blockwise mean of intensity was significantly smaller in the speakers with ASD compared with those with TD, although the SD of intensity within a whole session showed no significant difference between the two groups. The participants with TD had 8s blocks with both low and high intensity. This indicates that people with TD speak more frequently with loud or quiet voices depending on circumstances. In contrast, the individuals with ASD in our study spoke with a similar volume in each block.

For the SD of blockwise mean of log $F_0$, the variance among speakers was significantly greater in the ASD group than the TD group in spite of no significant difference for within-group means. The within-group SD of this feature was 82.6% larger in the ASD group compared with the TD group. The SD of log $F_0$ within a session had a similar (but not significant) tendency where the within-group variance was wider in the ASD group than in the TD group. These results are consistent with the finding by [17], which subdivided children with ASD into three groups with narrow, wide and typical ranges of $F_0$. Nevertheless, it is necessary to note that unlike that study, the SD was calculated among blocks in this study. The large SDs of blockwise mean of log $F_0$ are assumed to represent the large change in pitch among various contexts that appeared in the interaction with the conversational partner. From this point of view, the participants with ASD could be subdivided into groups that change their pitch in a wider, narrower, or typical range in different blocks. While some participants with ASD spoke in almost the same pitch in each block throughout the session, other participants changed pitch among blocks. The conflicting findings about the variance of $F_0$ among many previous researchers could be caused by the large distribution of the feature across individuals with ASD. Fusaroli *et al.* showed significantly wider $F_0$ variance in the population with ASD using meta-analysis, however, the speech data sets and the methods for measuring $F_0$ variance were diverse among studies, as the authors noted.

**Asynchrony of prosodic features.** The TD group had a higher correlation coefficient of the blockwise mean of intensity with their partner than the ASD group. A greater proportion of the participants with ASD had a negative correlation coefficient than those with TD. The results indicate that speakers with TD and their interlocutors spoke more synchronously than those with ASD and their partner. More participants with ASD spoke with quiet voices when the administrator spoke loudly, and vice versa. Synchrony is one of the components of entrainment related to social aspects such as speaker engagement [44]. As Gupta [22] showed, the degree of engagement will be different depending on circumstances in people with ASD as well. In this study, because the recorded conversations were limited to semi-structured interviews, the strength of synchrony could be compared among participants. Lower synchrony is possibly a characteristic of the conversation of people with ASD. However, the negative correlation coefficient of the acoustic features does not seem to simply mean disengagement: Pérez *et al.* showed that negative correlation coefficients also represent speaker engagement [21]. Further investigation should be conducted to identify the meaning of negative synchrony observed in some of the speakers with ASD.

**Long silence.** The individuals with ASD had significantly longer turn-taking gaps than those with TD. This finding was consistent with the results of Heeman *et al.* [18] who described long turn-taking gaps in children with ASD following a question, as in the ADOS activity used in the current study that consisted of questions and answers about emotional experiences. Furthermore, looking at each turn-taking gap within individuals, eight participants with ASD (12.9%) had one or more extended turn-taking gaps ($> 5$ s), which was not observed in any of the TD group. Speakers with TD inserted fillers, such as 'um' and 'uh' in

English, to stall for time and to keep their turn in a dialogue [45]. It is possible that individuals with ASD exhibiting long turn-taking gaps did not use fillers. The commonly-used English filler 'um' was observed significantly less frequently in children with ASD than those with TD, as reported by Clark *et al.* [46]. They proposed that 'um' is a signal of a lengthy upcoming delay common to speakers and listeners, based on comparison between 'um' and 'uh' by measuring listener comprehension [46]. In fact the delay after 'um' is longer than 'uh' but not as long as after 'uːm' [46]. Japanese fillers, their frequency of use and function, need to be investigated in future studies.

Individuals with ASD had significantly higher pause-to-turn ratios than those with TD. The ratio of the ASD group was 11% on average, three times that of the TD group. Morett *et al.* showed an elevated pause-to-turn ratio in children with ASD [47], and Thurber and Tager-Flusberg showed more frequent ungrammatical pauses in adolescents with ASD [48]. These findings about frequency were based on analysis of narrative production. In contrast, using interaction in ADOS administrations, Bone *et al.* found that intra-turn pauses correlated with ADOS scores [17]. It is considered from these findings that more frequent pauses were produced, and that silences were longer after taking a turn. People with ASD may allow more silence instead of inserting fillers or other utterances to keep their turn.

Turn-taking gaps of participants in the TD group were also characterized by a high frequency of negative gaps accompanying the responses initiated before the administrator's utterance ended. The mean relative frequency of negative turn-taking gaps was 61.7% in the TD group, while that frequency was 39.4% in ASD group. It is assumed that the negative turn-taking gaps also reduced the within-participant mean of turn-taking gaps. A previous multilingual comparison revealed that speakers generally minimize overlaps and turn-taking gaps, but Japanese speakers favor shorter gaps including negative ones [49]. These short gaps are conceivably related to the ability to predict turn ends, which can be predicted from lexico-syntactic and prosodic cues [50].

### Discrimination analysis

Discrimination analysis showed that selecting features revealed to be significantly different in *t*- or *F*-test (Setting D2) did not improve the accuracy of the discrimination. Selecting the best set of features provided 11.4-point better accuracy compared with Setting D2 (Table 4). However, the log mean of turn-taking gap and SD of blockwise mean of intensity, which showed significant differences in *t*-tests, greatly contributed to discrimination in Setting D3 (Table 5).

Focusing on a single speech feature, there was a significant overlap between the ASD and TD groups as shown in Fig 4. In addition, the combination of selected multiple speech features provided higher accuracy than when using one of them alone. Therefore, it is suggested that the conversational strategies of individuals with ASD can be characterized by a combination of multiple speech features. At the same time, it is possible that one or more speech features that serve as a clue for discrimination will differ in individuals or subgroups with ASD.

### Correlation between speech features and ADOS scores

Three speech features related to turn-taking and pausing had moderate correlations with the reciprocity score in ADOS (Table 6 and Fig 5). In ADOS, reciprocity was assessed by integrating speech and visual information used during social interaction, such as eye contact, facial expression and gesture. However, these aspects were not directly measured in this study. Strategies of turn-taking and pausing, together with physical behavior, reflect the mutual reciprocity between participant and administrator, and the characteristics of how the participant establishes and maintains communication. As shown in Fig 5, the speech features of

individuals with ASD who had low reciprocity scores were in almost the same range as the TD group. Speech features may not have decreased in the same proportion even in the case when the reciprocity score was close to zero.

In addition, for the SD of blockwise mean of intensity, individuals with features close to the mean of the TD group had low reciprocity scores, whereas those with features distant from the mean of TD had wide ranging scores (from 6 to 14). These observations indicate that correlation analysis is not necessarily sufficient to describe the relationship between speech features and ADOS scores. The prediction of ADOS score was degraded by the conversion of the blockwise mean of intensity using Eq (11), information about whether the feature played a role in the prediction.

## Regression analysis between speech features and ADOS score

The prediction of reciprocity scores using the features that showed a significant correlation with the ADOS (R2) score gave a better result than when all 13 features (R1), or the best features selected for the discrimination (R3), were used. Although Setting R3 was better than Setting R2, the correlation between predicted and actual scores was higher in Setting R4 than Setting R3 (Table 7). This means that the best combination of features for the classification does not provide the best performance in the regression.

In the selected features in Setting R4, the mean of speaking time gave the best prediction when it was used alone. However, it was the poorest predictor when the correlation-coefficient of blockwise mean intensity was omitted from the best set of features. This indicates that the correlation-coefficient of blockwise mean intensity improves the prediction performance when combined with other features.

As shown in Fig 6(a), two individuals with low actual reciprocity scores (3–5) received predictions that were too high, in error. This may have been because of the lack of speech samples from individuals with a low reciprocity score. As the performance of prediction of ADOS repetitive score was not sufficient, the speech features are considered not to depend on the characteristics related to repetitive score.

The mean of speaking time and correlation coefficient were selected for the prediction of both reciprocity and communication score. In contrast, speech rate was selected for communication score, not for reciprocity. The result relates to the criteria for communication scores in the ADOS protocol which includes assessment of speech rate.

## Potential limitations

The study has several limitations and methodological considerations. First, the number of participants with TD was small. Although the current findings were robust so that significance was maintained after correction for multiple comparisons, the detection of speech features related to ASD diagnosis could be limited. Second, race, sex, and age of the participants were limited to Japanese adult males. While this uniformity may have enhanced detection of speech features by controlling potential confounding effects of the participant characteristics, the findings should be applied to other populations with caution. Third, the estimation of IQs in TD and ASD groups was made in different methods. As IQ scores did not significantly correlated with the quantified speech features in the ASD or TD groups ($p>0.23$), it is less likely that the potential bias in the IQ assessment induced by the different methods had significant influences on the findings regarding differences in speech features between ASD and TD groups.

## Conclusions

In this study, conversation arising from a semi-structured interview was investigated to quantify the speech characteristics of people with ASD. We analyzed the speech features related to prosody, turn-taking, pausing, and synchrony of the conversation. Individuals with ASD spoke with a wide variety of pitch control strategies. The results also showed that individuals with ASD used similar volume regardless of the flow of conversation and with less synchronization with conversation partners, in comparison to those with TD. As regards to the timing in conversation, individuals with ASD responded after longer silence and more pauses after taking their turns. The discrimination analysis achieved high accuracy by combining the speech features related to the variance of prosody, duration of pause within speech, and duration of turn-taking gaps. The SD of blockwise mean of intensity contributed most to the discrimination.

The features of turn-taking and pausing correlated significantly with deficits associated with ASD in reciprocity. However, a simple correlation may not sufficiently describe the relationship between the speech features and deficits of ASD. Difficulties in reciprocity were effectively predicted using a set of features of SD and synchrony of intensity, mean of log duration of pause within speech, duration of turn-taking gap and log of pause-to-turn ratio. Synchrony of intensity contributed to the prediction of features of ASD even though it did not show a significant correlation with the ADOS score alone.

## Supporting information

**S1 Fig. Scatter plots representing the all calculated correlations between each speech feature and subscales of ADOS a) reciprocity, b) communication, and c) repetitive.** The (r) and p-value for correlation coefficient is shown in the top of each plot.
(PDF)

## Acknowledgments

Neither the funder nor sponsor, the Strategic Research Program for Brain Sciences from the Japan Agency for Medical Research and Development (JP18dm0107134), had any involvement in the data collection, analyses, writing, or interpretation of the study. The corresponding author had full access to the data and held the final responsibility regarding the decision to submit for publication. This work was also partially supported by a JSPS KAKENHI Grant-in-Aid for Scientific Research (A) (Grant Number: 16H01735). We thank Michelle Pascoe, PhD from Edanz Group (www.edanzediting.com/ac) for editing a draft of this manuscript.

## Author Contributions

**Conceptualization:** Hidenori Yamasue.

**Data curation:** Keiko Ochi, Miho Kuroda, Hidenori Yamasue.

**Formal analysis:** Keiko Ochi.

**Funding acquisition:** Nobutaka Ono, Hidenori Yamasue.

**Investigation:** Keiko Ochi, Keiho Owada, Masaki Kojima, Hidenori Yamasue.

**Methodology:** Keiko Ochi, Nobutaka Ono, Hidenori Yamasue.

**Supervision:** Nobutaka Ono, Shigeki Sagayama, Hidenori Yamasue.

**Writing – original draft:** Keiko Ochi, Hidenori Yamasue.

**Writing – review & editing:** Nobutaka Ono, Keiho Owada, Masaki Kojima, Miho Kuroda, Shigeki Sagayama, Hidenori Yamasue.

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
