## [Decision Letter · Decision Letter 0]

20 Aug 2019

PONE-D-19-20558

Quantization of speech and synchrony in the conversation of adults with autism spectrum disorder

PLOS ONE

Dear Dr. Yamasue,

Thank you for submitting your manuscript to PLOS ONE. After careful consideration, we feel that it has merit but does not fully meet PLOS ONE’s publication criteria as it currently stands. Therefore, we invite you to submit a revised version of the manuscript that addresses the points raised during the review process.

The reviewers addressed several concerns about your manuscript. Please revise your manuscript carefully.

We would appreciate receiving your revised manuscript by Oct 04 2019 11:59PM. To enhance the reproducibility of your results, we recommend that if applicable you deposit your laboratory protocols in protocols.io, where a protocol can be assigned its own identifier (DOI) such that it can be cited independently in the future. For instructions see: http://journals.plos.org/plosone/s/submission-guidelines#loc-laboratory-protocols

We look forward to receiving your revised manuscript.

Kind regards,

Kenji Hashimoto, PhD

Academic Editor

PLOS ONE

Journal Requirements:

Reviewers' comments:

Reviewer's Responses to Questions

**Comments to the Author**

1. Is the manuscript technically sound, and do the data support the conclusions?

Reviewer #1: Yes

Reviewer #2: Partly

2. Has the statistical analysis been performed appropriately and rigorously? 

Reviewer #1: Yes

Reviewer #2: No

3. Have the authors made all data underlying the findings in their manuscript fully available?

Reviewer #1: Yes

Reviewer #2: No

4. Is the manuscript presented in an intelligible fashion and written in standard English?

Reviewer #1: Yes

Reviewer #2: Yes

5. Review Comments to the Author

Reviewer #1: The authors shows that individuals with ASD speak with a uniform volume, uses wide variations in pitch depending on circumstances, be less synchronous with interlocutors in volume of voice, and longer pauses

between and within turns. Thus, authors mentioned that the length of pauses correlates with the reciprocity score in ADOS. This manuscript is valuable for publication.

Reviewer #2: In this manuscript, the authors investigated metrics extracted from speech-related features during a semi-structured socially interactive situations from the ADOS procedure, and evaluated the correlation between these metrics and the behavioral diagnosis of ASD.

Overall, the manuscript has a clearly defined research hypothesis. The experiments and analysis have been described in sufficient detail. Conclusions are drawn from the results that people with ASD showed quantitative difference in their speech-related features.

I have a few comments and concerns with regard to the analysis presented in this manuscript. There are a few places in the analysis that require further clarification.

1. The assessment of IQs in TD and ASD groups was made in different methods. The authors should discuss/rule-out the possible bias in the IQ assessment induced by the different methods for TD and ASD groups.

2. Ln 252-254, for the calculation of turn-taking gaps, the authors omitted utterances that overlapped entirely with the administrator's utterance. This may induce bias towards larger turn-taking gaps. The authors should discuss/analyze the possible bias.

3. Ln 277, how are the data weighted in the SVM? Weighting the data points in the SVM cost function would affect the classification results, especially the trade-off between type-I and type-II errors. One possible solution is to perform ROC analysis of the SVM and report the full ROC curve or AUC of the ROC curve.

4. Ln 301, taking absolute value to transform the speech-related features would affect the distribution of these features (no longer normally distributed), hence violating the gaussian assumption in Pearson's correlation if the p-values in the correlation analysis were computed from a t-distribution. In this case, the authors should consider non-parametric test (e.g. permutation test) for correlation analysis.

5. Figure 4 and Table 2 described the differences between TD and ASD groups qualitatively and quantitatively. Some of the features are clearly non-gaussian, and the central limit theorem may not be applicable here since the number of samples in TD is relatively small. The authors may consider performing permutation test to evaluate the statistical significance of these differences.

6. The x axis tick labels in Figure 4 is confusing, it might be better just include fewer ticks. E.g. just use 0, 0.1, 0.2, 0.3 in panel (a) instead of writing down all the bins explicitly.

7. Setting D3 was described as "the best combination of features selected based on accuracy from all possible 8191 combinations". It is not clear whether the authors use an independent validation/testing set to estimate the performance using D3. If not, then the results in Table 3 and 4 may suffer from overfitting.

8. Similarly, setting R4 also used the best combination of features. Results in Table 7 and Figure 6 may suffer from overfitting. The authors should describe whether cross-validation or independent validation/testing set was used.

9. It might be better to add the corresponding r and p values to Figure 5 and 6 for better presentation of the results.

10. Ln 418-419, the author should consider adding scatter plot for all the correlation analysis, not only of the significant features, but also the insignificant ones. Maybe the insignificant ones can be included in supplement figures.

11. The authors should add the type of scores to the title in each of the 3 panels in Figure 6 to make it clear.

6. PLOS authors have the option to publish the peer review history of their article (what does this mean?). If published, this will include your full peer review and any attached files.

Reviewer #1: No

Reviewer #2: No

---

## [Author Response · Author response to Decision Letter 0]

15 Oct 2019

Thank you for your kind letter dated August 21, 2019, regarding manuscript PONE-D-19-20558, entitled “Quantization of speech and synchrony in the conversation of adults with autism spectrum disorder”. We greatly appreciate the helpful comments and suggestions for revision provided by the editor and reviewers.

We have substantially revised the manuscript in accord with the reviewers’ suggestions. Our point-by-point responses to each of the reviewers’ comments are presented below. The revised sections of the text are underlined in the manuscript including other minor changes (correcting typo, etc.).

Reviewer #1: The authors shows that individuals with ASD speak with a uniform volume, uses wide variations in pitch depending on circumstances, be less synchronous with interlocutors in volume of voice, and longer pauses between and within turns. Thus, authors mentioned that the length of pauses correlates with the reciprocity score in ADOS. This manuscript is valuable for publication.

We very much appreciate the reviewer’s sound and kind review.

Reviewer #2: In this manuscript, the authors investigated metrics extracted from speech-related features during a semi-structured socially interactive situations from the ADOS procedure, and evaluated the correlation between these metrics and the behavioral diagnosis of ASD.

 Overall, the manuscript has a clearly defined research hypothesis. The experiments and analysis have been described in sufficient detail. Conclusions are drawn from the results that people with ASD showed quantitative difference in their speech-related features.

 I have a few comments and concerns with regard to the analysis presented in this manuscript. There are a few places in the analysis that require further clarification.

We also very much appreciate the reviewer’s careful review and helpful comments.

1. The assessment of IQs in TD and ASD groups was made in different methods. The authors should discuss/rule-out the possible bias in the IQ assessment induced by the different methods for TD and ASD groups.

We thank the reviewer’s insightful comment. As IQ scores did not significantly correlated with the quantified speech features in the ASD or TD groups (p>0.23), it is less likely that the potential bias in the IQ assessment induced by the different methods had significant influences on the findings regarding differences in speech features between ASD and TD groups. However, as the reviewer pointed, we cannot totally rule out the possible bias. Therefore, we added a brief discussion on this issue in the limitation section of the revised manuscript as follows:

In line 638-643 of the revised manuscript “Third, the estimation of IQs in TD and ASD groups was made in different methods. As IQ scores did not significantly correlated with the quantified speech features in the ASD or TD groups (p>0.23), it is less likely that the potential bias in the IQ assessment induced by the different methods had significant influences on the findings regarding differences in speech features between ASD and TD groups.”

2. Ln 252-254, for the calculation of turn-taking gaps, the authors omitted utterances that overlapped entirely with the administrator's utterance. This may induce bias towards larger turn-taking gaps. The authors should discuss/analyze the possible bias. 

We consider that the utterances that overlapped entirely with the administrator's utterance can be roughly classified into two types. One is a backchannel. As many studies define the backchannels as inserted utterances without interrupting the interlocutor's speech (Heldner, et al., 2011), most of the participants' fully overlapped utterances are considered to be backchannels. The other is a failure of taking turns, which means that the participant continued to speak even after the administrator tried to take a turn. As regards to a backchannel, we did not consider them as a turn-taking as the study by Sato et al. (2002) did. We also considered that the failure of turn-taking are not included in a new turn. Thus, the bias caused by omitting the fully overlapped utterances are considered to be little.

Based on the above discussions, we changed the manuscript in lines 252-254 “For the calculation of turn-taking gaps, we omitted utterances that overlapped entirely with the administrator’s utterance, regarding them as backchannels” to “We did not include the backchannels, which are defined as the utterances inserted into the interlocutor’s utterances without interrupting in many literatures [38], into a turn based on the study by Sato et al. [39]. For the calculation of turn-taking gaps, we omitted utterances that overlapped entirely with the administrator’s utterance, regarding them as backchannels or failures of turn-taking in which the participant continued to speak after the administrator tried to take a turn.” in lines 253-258 of the revised manuscript. With the change of the manuscript, we added the reference [38] and [39] in the revised manuscript.

[38] Heldner M, Edlund J. Pauses, Gaps and Overlaps in Conversations. Journal of Phonetics. 2010;38(4):555-568.

[39] Sato R, Higashinaka R, Tamoto M, Nakano M, Aikawa K. Learning Decision Trees to Determine Turn-Taking by Spoken Dialogue Systems. Proceedings of 7th International Conference on Spoken Language Processing. 2002;861-864.

3. Ln 277, how are the data weighted in the SVM? Weighting the data points in the SVM cost function would affect the classification results, especially the trade-off between type-I and type-II errors. One possible solution is to perform ROC analysis of the SVM and report the full ROC curve or AUC of the ROC curve.

In the training of SVM, we weighted the objective function according to the inverse number of the samples of each class (Rosenberg, 2012). For clarifying this, we modified the following manuscripts in the lines 277-278: “The training data for the ASD and TD groups were appropriately weighted to compensate for the differences in data size.” into “Because the sample sizes of the two groups were imbalanced, we weighted the objective function used in the training of the SVM according to the inverse number of the sample size of each class to avoid the undervaluation of the false negative.” in lines 288-291 of the revised manuscript.

4. Ln 301, taking absolute value to transform the speech-related features would affect the distribution of these features (no longer normally distributed), hence violating the gaussian assumption in Pearson's correlation if the p-values in the correlation analysis were computed from a t-distribution. In this case, the authors should consider non-parametric test (e.g. permutation test) for correlation analysis.

We thank the reviewer’s insightful comment. Based on the reviewer’s comment, we used permutation test as a non-parametric correlational analysis for the SD of blockwise mean of log F0 that was analyzed after the transform using Equation (10). The non-normality of the transformed feature was shown from the Kolmogorov-Smirnov test (p < 0.05). Please also see the answer to comment 5 about details. We added the manuscript in the lines 320-322 of the revised manuscript “We used permutation test for the SD of blockwise mean of log F0, which did not have a normal distribution (Kolmogorov-Smirnov test, p < 0.05).” We also modified Table 6.

The modified Table 6 includes the revision related to the answer for the question 3 (for details, see answer to comment 5.

5. Figure 4 and Table 2 described the differences between TD and ASD groups qualitatively and quantitatively. Some of the features are clearly non-gaussian, and the central limit theorem may not be applicable here since the number of samples in TD is relatively small. The authors may consider performing permutation test to evaluate the statistical significance of these differences.

We thank the reviewer’s insightful comment. In response to the comment, we conducted Kolmogorov-Smirnov tests on each speech features in the ASD and TD groups to investigate the non-normality. The results showed significance on the distributions of the mean and SD of turn-taking gap and turn-to-pause ratio of the TD group. Based on these results, we performed logarithm transform on these three speech features. For only the mean of turn-taking gap, we added a constant to all samples such that the minimum value of the feature was equal to 1 prior to the logarithm transform for non-negativity. Then, we confirmed that Kolmogorov-Smirnov test showed no significance in all of these three speech features. This insignificance was consistent with the results of Weilmhammer et al. (2003) and Bosch et al (2004) that showed the normality of the distribution of the logarithm of durations of turn-taking gaps and overlaps. Therefore, we added manuscript “To assess the normality of the distribution of the speech features of the ASD and TD group, we conducted Kolmogorov-Smirnov tests on each feature of each group. Based on the results which showed the non-normality in the distribution of the mean and SD of turn-taking gap and the turn-to-pause ratio (p < 0.05), we took the logarithm transform on the three features, referring to the findings that the logarithm of the duration of the gaps and overlaps of turn takings have a normal distribution [40] [41]. For only the mean of turn-taking gap, we added a constant to all samples such that the minimum value of the feature was equal to 1 prior to the logarithm transform for non-negativity. We confirmed that Kolmogorov-Smirnov test showed no significance on these logarithmically transformed features.” in the lines 271-279 of the revised manuscript. We also modified Table 2 according to the result of t- and F-tests of the transformed features.

Because the experimental results are slightly changed by the logarithm transform to three speech features, we revised several related sentences. We listed them as follows. FS and R1 denote the sentences in the first submission and the first revision (this submission)

[FS] lines 339-341 “Both t- and F-tests showed significant differences between the ASD and TD group in the mean of turn-taking gap, the SD of turn-taking gap, and the pause-to-turn ratio (adjusted p<0.05) (Table 2, Figure 5).”

[R1] lines 366-368 “Both t- and F-tests showed significant differences between the ASD and TD group in the log mean of turn-taking gap, and the log SD of turn-taking gap (adjusted p<0.05) (Table 2, Figure 5)”

[FS] We also changed the manuscript in lines 358-360 “Regarding the SD and correlation coefficient of blockwise mean of intensity, the t-test found significant differences between ASD and TD groups, although the F-test found no significant difference.” 

[R1] lines 385-387 “Regarding the SD and correlation coefficient of blockwise mean of intensity, the pause-to-turn ratio, the t-test found significant differences between ASD and TD groups, although the F-test found no significant difference.”

 We also added the manuscript “The pause-to-turn ratio of those of ASD was significantly larger than that of ASD.” in the line 397.

 We deleted the manuscript in lines 288-292 of the original manuscript “In the correlational analyses, logarithms were taken of the mean speaking time, SD of speaking time, SD of turn-taking gaps and pause-to-turn ratio, because the samples were concentrated on the small values of these features. Although the mean of turn-taking gaps had a distribution concentrated in the small value, a logarithm was not used because it could take a negative value.”

Along with the logarithmic transforms before t- and F-tests mentioned above, we also applied the same shift and logarithm transformation on the mean of turn-taking gap in the subsequent discrimination, correlational and regression analysis. Based on these revision, we changed the following manuscripts. The name of the feature “mean of turn-taking gap” was changed to “log mean of turn-taking gap” in the revised manuscript in these steps. With the revision, the results of discrimination analysis shown in Table 3, 4 and 5 was modified.

[FS] in lines 385-387 “The mean of turn-taking gap and the SD of blockwise mean of intensity were included in at least the top 100 sets with accuracy, from all 8191 combinations.” 

[R1] in lines 414-416 “The log mean of turn-taking gap and the SD of blockwise mean of intensity were included in at least the top 20 sets with accuracy, from all 8191 combinations.”

Table 6 was also modified following the conversion of the mean of turn-taking gap as shown in the answer to comment 5. The numbers of the accuracies and F-measures shown in lines 380-382 were modified according to the result in Table 4-6. 88.6%, 93.4% and 20.3% were modified to 89.9%, 93,4% and 20.3%, respectively. The value of the correlation coefficient in line 401 (r = 0.44) was modified to (r = 0.41) following the Table 6.

[FS] in lines 438-440 “The best set of features included the correlation coefficient of the blockwise mean of intensity and log of mean of speaking time for the prediction of reciprocity and communication score.”

[R1] in lines 466-468 “The speech features selected for the prediction of communication and repetitive score was respectively involved in the selected features for the prediction of reciprocity.” 

[FS] in lines 442-446 “Using only the correlation coefficient of blockwise mean of intensity alone, the predicted reciprocity score did not yield a positive correlation with the rated score. However, the correlation between rated and predicted score was most degraded (0.43 from 0.56) when the feature was excluded from the best feature set. The top-21 feature sets for the prediction of reciprocity score included the correlation coefficient of blockwise mean of intensity.” 

[R1] in lines 470-473 ”Using only the correlation coefficient of blockwise mean of intensity alone, the predicted reciprocity score did not yield a correlation with the rated score. However, the correlation between rated and predicted score was the third most degraded (0.47 from 0.58) when the feature was excluded from the best feature set.”

Additionally, we added the following references:

[40] Weilhammer K, Rabold S. Durational Aspecats in Turn Taking. Proceedings of the International Conference of Phonetic Sciences. 2003.

[41] Ten Bosch L, Oostdijk N, De Ruiter JP. Durational Aspects of Turn-Taking in Spontaneous Face-to-Face and Telephone Dialogues. Proceedings of International Conference on Text, Speech and Dialogue. 2004; 563-570.

6. The x axis tick labels in Figure 4 is confusing, it might be better just include fewer ticks. E.g. just use 0, 0.1, 0.2, 0.3 in panel (a) instead of writing down all the bins explicitly.

We thank the reviewer’s careful review, and modified the bins of Figure 4 according to the reviewer’s comment.

7. Setting D3 was described as "the best combination of features selected based on accuracy from all possible 8191 combinations". It is not clear whether the authors use an independent validation/testing set to estimate the performance using D3. If not, then the results in Table 3 and 4 may suffer from overfitting.

“The possible 8191 combinations” mean: (Please see the response letter)

which represents the number of all cases where each of 13 speech features were used or not.

 In evaluation of the D3 setting, we used leave-one-out cross-validation to avoid overfitting. In details, the accuracy of each combination of speech features (Combination i) was evaluated by the following equation: (Please see the response letter)

where takes 1 or 0 when the classification is accurate or not, respectively, in the test for the kth participant’s data by the classifier trained using all the other participants’ data (i.e., all 78 participants except for the kth one). represents the total number of the participants of the ASD and TD groups (79 in this study). 

 Based the above explanations, we added the following manuscript to the lines 295-304 of the revised manuscript: “The number of the possible combination (8191) represents the number of all cases where each of 13 speech features were used or not (). In the evaluation of the D3 setting, we used leave-one-out cross-validation to avoid overfitting. In details, the accuracy of each combinations of speech features (Combination i) was evaluated by the following equation: (Please see the response letter)

where takes 1 or 0 when the classification is accurate or not respectively, in the test for the kth participant’s data by the classifier trained using all the other participants’ data (i.e., all 78 participants except for the kth one). represents the total number of the participants of the ASD and TD groups (79 in this study). ”

8. Similarly, setting R4 also used the best combination of features. Results in Table 7 and Figure 6 may suffer from overfitting. The authors should describe whether cross-validation or independent validation/testing set was used.

Similar to the answer to the Comment 7, the number of the possible combination was calculated. The leave-one-out cross validation was carried out in the same way, and the correlation coefficients, MAE, and RMSE in Table 7 were obtained by averaging the values when the data a single participant was tested from the regressor trained by the data of the other participants, respectively. Therefore, we added the following manuscript into the lines 334-338 of the revised manuscript: “Similar to the Setting D3 in the discrimination analysis, the possible 8191 combination was evaluated in Setting R4. Carrying out one-leave-out cross-validation in the same way, the correlation coefficients, MAE, and RMSE were obtained by averaging the values when the data a single participant was tested from the regressor trained by the data of the other participants, respectively.”

9. It might be better to add the corresponding r and p values to Figure 5 and 6 for better presentation of the results.

In response to the comment, we added the values into the Figure 5 and 6 following to the comments.

10. Ln 418-419, the author should consider adding scatter plot for all the correlation analysis, not only of the significant features, but also the insignificant ones. Maybe the insignificant ones can be included in supplement figures.

Based on the suggestion, we added the scatter plots of the all correlation analyses into the supporting information section.

11. The authors should add the type of scores to the title in each of the 3 panels in Figure 6 to make it clear.

Following to the reviewer’s comment, we added the title which represents the type of scores into Figure 6.

We believe that the manuscript has been substantially improved by incorporating the many helpful comments from the reviewers, and we deeply appreciate their valuable input. Please do not hesitate to contact us if any further information is required.

---

## [Decision Letter · Decision Letter 1]

31 Oct 2019

PONE-D-19-20558R1

Quantization of speech and synchrony in the conversation of adults with autism spectrum disorder

PLOS ONE

Dear Dr. Yamasue,

Thank you for submitting your manuscript to PLOS ONE. After careful consideration, we feel that it has merit but does not fully meet PLOS ONE’s publication criteria as it currently stands. Therefore, we invite you to submit a revised version of the manuscript that addresses the points raised during the review process.

The reviewer addressed some minor concerns about your revised manuscript. Please revise your manuscript carefully.

We would appreciate receiving your revised manuscript by Dec 15 2019 11:59PM. To enhance the reproducibility of your results, we recommend that if applicable you deposit your laboratory protocols in protocols.io, where a protocol can be assigned its own identifier (DOI) such that it can be cited independently in the future. For instructions see: http://journals.plos.org/plosone/s/submission-guidelines#loc-laboratory-protocols

We look forward to receiving your revised manuscript.

Kind regards,

Kenji Hashimoto, PhD

Academic Editor

PLOS ONE

Reviewers' comments:

Reviewer's Responses to Questions

**Comments to the Author**

1. If the authors have adequately addressed your comments raised in a previous round of review and you feel that this manuscript is now acceptable for publication, you may indicate that here to bypass the “Comments to the Author” section, enter your conflict of interest statement in the “Confidential to Editor” section, and submit your "Accept" recommendation.

Reviewer #1: All comments have been addressed

Reviewer #2: (No Response)

2. Is the manuscript technically sound, and do the data support the conclusions?

Reviewer #1: Yes

Reviewer #2: Yes

3. Has the statistical analysis been performed appropriately and rigorously? 

Reviewer #1: Yes

Reviewer #2: Yes

4. Have the authors made all data underlying the findings in their manuscript fully available?

Reviewer #1: Yes

Reviewer #2: Yes

5. Is the manuscript presented in an intelligible fashion and written in standard English?

Reviewer #1: Yes

Reviewer #2: Yes

6. Review Comments to the Author

Reviewer #1: Authors have responded to the original concerns raised by the reviewers, and the quality of the manuscript has significantly improved.

Reviewer #2: The authors have adequately addressed my comments. I only have some minor points:

1. Line 368, the authors wrote “(Table 2, Figure 5)”. Are they referring to “Table 2, Figure 4”?

2. Line 583, “…as shown in Figure 5.” Should it also be “… as shown in Figure 4”?

3. Figure 5 seems confusing. The negative r values apparently did not match the positive regression slope.

7. PLOS authors have the option to publish the peer review history of their article (what does this mean?). If published, this will include your full peer review and any attached files.

Reviewer #1: No

Reviewer #2: No

---

## [Author Response · Author response to Decision Letter 1]

1 Nov 2019

Thank you for your kind letter dated October 31, 2019, regarding manuscript PONE-D-19-20558R1, entitled “Quantization of speech and synchrony in the conversation of adults with autism spectrum disorder”. We greatly appreciate the helpful comments and suggestions for revision provided by the editor and reviewer.

We have substantially revised the manuscript in accord with the reviewers’ suggestions. Our point-by-point responses to each of the reviewers’ comments are presented below. The revised sections of the text are underlined in the manuscript.

 Reviewer #2: The authors have adequately addressed my comments. I only have some minor points: 

We very much appreciate the reviewer’s careful reading.

 

Comment 1. Line 368, the authors wrote “(Table 2, Figure 5)”. Are they referring to “Table 2, Figure 4”?

 2. Line 583, “…as shown in Figure 5.” Should it also be “… as shown in Figure 4”?

Response: We carefully confirmed these parts, and corrected these typos.

 

Comment 3. Figure 5 seems confusing. The negative r values apparently did not match the positive regression slope.

Response: Thank you for the reviewer’s careful review. Based on the comment, we noticed that the r and p values for the regressions with ADOS repetitive score were incorrectly presented as those with ADOS reciprocity score in the Figure 5. Now, the r and p values for the regressions with ADOS reciprocity score were correctly presented in the revised Figure 5.

In addition, we would like to add a minor change in the title of this paper. We used the word "quantization" to express the conversion of the continuous values such as speech or conversational synchrony to discrete ADOS scores, however, we noticed that "quantization" was confusing because sometimes it is utilized to express the digitalizing of sound waveforms. Therefore, we would like to change the word "quantization" into "quantification" as follows:

The original title: “Quantization of speech and synchrony in the conversation of adults with autism spectrum disorder”

The revised one: “Quantification of speech and synchrony in the conversation of adults with autism spectrum disorder”

We believe that the manuscript has been substantially improved by incorporating the many helpful comments from the reviewers, and we deeply appreciate their valuable input. Please do not hesitate to contact us if any further information is required.

---

## [Editor Report · Decision Letter 2]

5 Nov 2019

Quantification of speech and synchrony in the conversation of adults with autism spectrum disorder

PONE-D-19-20558R2

Dear Dr. Yamasue,

We are pleased to inform you that your manuscript has been judged scientifically suitable for publication and will be formally accepted for publication once it complies with all outstanding technical requirements.

With kind regards,

Kenji Hashimoto, PhD

Section Editor

PLOS ONE
---

## [Editor Report · Acceptance letter]

8 Nov 2019

PONE-D-19-20558R2 

Quantification of speech and synchrony in the conversation of adults with autism spectrum disorder 

Dear Dr. Yamasue:

I am pleased to inform you that your manuscript has been deemed suitable for publication in PLOS ONE. Congratulations! Your manuscript is now with our production department. 

With kind regards,

on behalf of

Prof. Kenji Hashimoto 

Section Editor

PLOS ONE